# Reconstitution of EBV-directed T cell immunity by adoptive transfer of peptide-stimulated T cells in a patient after allogeneic stem cell transplantation for AITL

María Fernanda Lammoglia Cobo[1⊚], Julia Ritter[2⊚], Regina Gary[3], Volkhard Seitz[2,4], Josef Mautner[5,6], Michael Aigner[3], Simon Völkl[3], Stefanie Schaffer[3], Stephanie Moi[3], Anke Seegebarth[2], Heiko Bruns[3], Wolf Rösler[3], Kerstin Amann[7], Maike Büttner-Herold[7], Steffen Hennig[4], Andreas Mackensen[3], Michael Hummel[2], Andreas Moosmann[5,6‡], Armin Gerbitz[8‡*]

1 Department of Hematology, Oncology, and Tumor Immunology, Charité–Universitätsmedizin Berlin, corporate member of Freie Universität Berlin and Humboldt-Universität zu Berlin, Berlin, Germany, 2 Institute of Pathology, Charité–Universitätsmedizin Berlin, corporate member of Freie Universität Berlin and Humboldt-Universität zu Berlin, Berlin, Germany, 3 Department of Internal Medicine 5 –Hematology/ Oncology, University Hospital Erlangen, Erlangen, Germany, 4 HS Diagnomics GmbH, Berlin, Germany, 5 Department of Medicine III, LMU-Klinikum, Munich, Germany, 6 German Centre for Infection Research, Munich, Germany, 7 Department of Nephropathology, Institute of Pathology, University of Erlangen, Erlangen, Germany, 8 Division of Medical Oncology and Hematology, Princess Margaret Cancer Center, Toronto, Ontario, Canada

⊚ These authors contributed equally to this work.
‡ AM and AG also contributed equally to this work.
* armin.gerbitz@uhn.ca

**Data Availability Statement:** All relevant data are within the manuscript and its Supporting Information files.

## Abstract

Reconstitution of the T cell repertoire after allogeneic stem cell transplantation is a long and often incomplete process. As a result, reactivation of Epstein-Barr virus (EBV) is a frequent complication that may be treated by adoptive transfer of donor-derived EBV-specific T cells. We generated donor-derived EBV-specific T cells by stimulation with peptides representing defined epitopes covering multiple HLA restrictions. T cells were adoptively transferred to a patient who had developed persisting high titers of EBV after allogeneic stem cell transplantation for angioimmunoblastic T-cell lymphoma (AITL). T cell receptor beta (TCRβ) deep sequencing showed that the T cell repertoire of the patient early after transplantation (day 60) was strongly reduced and only very low numbers of EBV-specific T cells were detectable. Manufacturing and *in vitro* expansion of donor-derived EBV-specific T cells resulted in enrichment of EBV epitope-specific, HLA-restricted T cells. Monitoring of T cell clonotypes at a molecular level after adoptive transfer revealed that the dominant TCR sequences from peptide-stimulated T cells persisted long-term and established an EBV-specific TCR clonotype repertoire in the host, with many of the EBV-specific TCRs present in the donor. This reconstituted repertoire was associated with immunological control of EBV and with lack of further AITL relapse.

**Funding:** AG was supported by the ZIM fund (KF 2766401FRO, Federal Ministry for Economic Affairs and Energy Germany), the BayImmuNet Consortium of the Bayerische Staatsministerium für Bildung und Kultur, Wissenschaft und Kunst (Government of Bavaria), and the Deutsche Forschungsgemeinschaft (DFG, German Research Foundation) - Projektnummer 324392634 - TRR 221 and SFB643. MFLC was supported by scholarships from Berlin School of Integrative Oncology (BSIO), Charité – Universitätsmedizin Berlin, and the Mexican National Council of Science and Technology (CONACYT). AMo was supported by Wilhelm Sander-Stiftung (project 2018.135.1). The funders had no role in study design, data collection and analysis, decision to publish, or preparation of the manuscript.

**Competing interests:** I have read the journal's policy and the authors of this manuscript have the following competing interests: AMo receives project funding from Biosyngen Pte. Ltd. for preclinical development of EBV-specific TCRs not related to this study.

## Author summary

A characteristic feature of all herpesviruses is their persistence in the host's body after primary infection. Hence, the host's immune system is confronted with the problem to control these viruses life-long. When the immune system is severely compromised, for example after stem cell transplantation from a foreign (allogeneic) donor, these viruses can reappear, as they persist in the host's body life-long after primary infection. Epstein-Barr virus (EBV) is a herpesvirus that can cause life-threatening complications after stem cell transplantation and only reinforcement of the host's immune system can reestablish control over the virus. Here we show that *ex vivo* manufactured EBV-specific T cells can reestablish long-term control of EBV and that these cells persist in the host's body over months. These results give us a better understanding of viral immune reconstitution post-transplant and of clinically-relevant T cell populations against EBV.

## Introduction

Linked to its high prevalence in adults, approximately 30–40% of patients reactivate Epstein-Barr virus (EBV) after MHC-matched allogeneic stem cell transplantation (allo-SCT), as determined by virus-specific PCR of cells of the peripheral blood [1]. Reactivation of EBV worsens outcome after allo-SCT, since it imposes the risk of EBV-associated post-transplant lympho-proliferative disorder (PTLD) and is associated with malignancies such as angioimmunoblastic T cell lymphoma (AITL) [2]. AITL is a rare form of T cell non-Hodgkin Lymphoma in which concomitant EBV infection often occurs [3]. EBV appears to play a role in AITL pathogenesis and histological development [2,4], either through EBV-infected B immunoblasts found at early AITL stages adjacent to neoplastic T cells [5–7] or infection of both cells types [8]. For this reason, EBV serostatus and viral loads serve as important prognostic factors [9,10], especially among young patients [11].

EBV DNA load in peripheral blood is routinely monitored by polymerase chain reaction (PCR) in patients after allo-SCT to allow for pre-emptive treatment strategies [12]. Since no specific antiviral therapy is available to date, treatment of EBV-related disease in patients after allo-SCT focuses on three major strategies: (i) in-patient depletion of EBV-transformed B cells with antibodies -with depletion of other B cells as collateral damage- (ii) reduction of immuno-suppression, or (iii) application of EBV-specific, donor-derived T cells [13–16].

The availability of B cell-depleting antibodies has reduced the occurrence of PTLD after allo-SCT [17], but comes with severe side effects and costs. Due to the long-term depletion of B cells, antibody generation is abolished and patients are at risk of severe infections, especially with encapsulated bacteria whose control requires antibody opsonization [18,19]. Therefore, frequent application of intravenous immunoglobulins is necessary. Furthermore, the problem of failing immunological control of EBV is not resolved.

As an alternative strategy, several groups have focused on the development of EBV-specific T cell transfer, as reactivation of EBV is associated with use of T cell-depleted grafts or insufficient T cell reconstitution after transplantation. This approach does not bear the risk of developing *de novo* graft versus host disease [20–24]. Adoptive transfer of natural EBV-specific T cells from EBV-positive donors has been performed and is considered overall a success due to its effectiveness and safety [23]. For patients with EBV-seronegative donors, where natural EBV-specific T cells are not available, adoptive transfer of EBV TCR-transduced T cells is a promising alternative [25–29].

We have recently described a Good Manufacturing Practice (GMP)-compliant method for the generation of CMV- and EBV-specific T cells by stimulation of G-CSF mobilized allogeneic stem cell grafts or conventional PBMC with MHC-I- and MHC-II-restricted epitope peptides derived from viral proteins [24]. We selected peptides that allow for comprehensive quality control of the product and subsequent follow-up within the patient after adoptive transfer, using flow cytometry with peptide-MHC multimers. However, little is known about the detailed structure of the EBV-specific T cell repertoire recognizing each epitope and its fate after adoptive transfer into the patient.

Here, we generated multi-epitope-specific T cells by peptide stimulation and adoptively transferred them to a patient with severe EBV reactivation after allo-SCT. We selected several peptides for defined latent and lytic epitopes of multiple well-established HLA restrictions. Using peptide-MHC multimer binding in flow cytometry and high-throughput sequencing of the TCRβ repertoire, we show that stimulation of T cells with EBV peptides generates a product with a clonotypic TCRβ repertoire that is strongly focused on EBV-specific sequences, and that this repertoire can be tracked long-term *in vivo* after adoptive transfer in the patient to demonstrate immune reconstitution.

## Results

### Manufacturing of EBV-specific T cells

A 55-year-old, EBV-seropositive patient suffered from biopsy-confirmed chemotherapy-refractory AITL (Stage IVB, EBV⁻, see S1 and S2 Figs) and was transplanted with G-CSF-mobilized peripheral blood stem cells from an HLA 10/10 matched unrelated donor. Concomitant with leukemic relapse shortly after transplantation (day 42), the patient developed high EBV titers in peripheral blood on day 66 and received conventional unmanipulated donor lymphocyte infusion (DLI) on day 76 and Rituximab four times weekly on days 68–89. As major symptoms of EBV reactivation (night sweats, fever, itching of the skin, and elevated liver enzymes) were not controlled, we decided to prepare and adoptively transfer peptide-stimulated EBV-specific T cells (ATCT) on day 105. An overview of the patient history is provided in S1 Table and S3 Fig.

To prepare EBV-specific T cells, a total of 600 million conventional PBMC (frozen fraction of the preparation for conventional DLI) were stimulated with a pool of defined EBV-derived peptides (1 μg/ml per peptide, Table 1), similar to the procedure published previously [24]. Peptide-stimulated cells were subsequently expanded in a closed bag system for 9 days. Fig 1A (left panel) shows the composition of the PBMC before peptide stimulation (day 0) and of the resulting cell composition after 9 days of expansion. The dominant fraction of cells in the product were CD3⁺ T cells (84.8%). B cells, NK cells and monocytes were reduced to 5.8% of all cells. Other cells (9.4%) were mainly macrophages, activated monocytes, neutrophils (all CD11b⁺, CD68⁺), and few remaining granulocytes. As shown in Fig 1A (right panel), total CD3⁺ T cell number increased from approximately 315 million to 631 million cells over the 9-day period, and a total of approximately 750 million cells were harvested.

The T cell product was analyzed before and after peptide stimulation with peptide-MHC multimers (Fig 1B) corresponding to the six peptides of the stimulation pool that were restricted through HLAs present in transplant donor and recipient (Table 1). On day 0, 2.4% of CD8⁺ T cells, mainly in the CCR7-negative subset, specifically bound peptide-MHC multimers. By day 9, T cells specific for five of the six epitopes had strongly expanded and now amounted to 64.6% of all CD8⁺ T cells. Two epitopes (RAK and EPL) from the immediate-early protein BZLF1 and one epitope (HPV) from the latent antigen EBNA1 were particularly dominant. Intracellular cytokine staining after restimulation demonstrated, as expected [44],

**Table 1. Peptide pool used for T cell stimulation.**

| label | AA Sequence | peptide length | protein | presented on HLA | reference | matched with patient |
|-------|-------------|----------------|---------|------------------|-----------|----------------------|
| CLG | CLGGLLTMV | 9 | LMP2 | A*02:01 | [30] | |
| GLC | GLCTLVAML | 9 | BMLF1 | A*02:01 | [31,32] | |
| YVL | YVLDHLIVV | 9 | BRLF1 | A*02:01 | [33] | |
| FLY | FLYALALLL | 9 | LMP2 | A*02:01 | [34] | |
| RLR | RLRAEAQVK | 9 | EBNA3A | A*03:01 | [35] | + |
| RPP | RPPIFIRRL | 9 | EBNA3A | B*07:02 | [35] | |
| QAK | QAKWRLQTL | 9 | EBNA3A | B*08:01 | [36] | + |
| RAK | RAKFKQLL | 8 | BZLF1 | B*08:01 | [37] | + |
| YPL | YPLHEQHGM | 9 | EBNA3A | B*35:01 | [36] | + |
| HPV | HPVGEADYFEY | 11 | EBNA1 | B*35:01 | [38] | + |
| EPL | EPLPQGQLTAY | 11 | BZLF1 | B*35:01 | [33] | + |
| PYYV | PYYVVDLSVRGM | 12 | BHRF1 | DR*4 | [39] | |
| VVRM | VVRMFMRERQLPQS | 14 | EBNA3C | DR*11 | [40] | |
| FGQL | FGQLTPHTKAVYQPR | 15 | BLLF1 | DR*13 | [41] | |
| IPQC | IPQCRLTPLSRLPFG | 15 | EBNA1 | DR*13 | [42] | |
| TDAW | TDAWRFAMNYPRNPT | 15 | BNRF1 | DR*15 | [43] | |

AA: amino acid sequence.

that a variable proportion of CD8+ T cells specific for these epitopes secreted IFN-γ in response to single peptide stimulation (Fig 1C). A high proportion of IFN-γ-secreting CD8+ cells (13.2%, compared to 24.1% of multimer-staining cells) was seen for the EPL epitope. An increase of IFN-γ concentration and other cytokines was also detected in patient serum after adoptive transfer (S4 Fig). While more than half (53.6%) of CD3+ T cells of the PBMCs (day 0) had initially a naive phenotype (CCR7+/CD45RA+), these were reduced to 15.4% on day 9 (Fig 1D). In contrast, the proportion of T cells with effector/effector memory phenotype (CCR7-/CD45RA-) changed from 22.8% to 80.0%. The three most dominant multimer-binding CD8+ T cell populations (EPL, RAK, and HPV) had a dominating effector memory phenotype in the cellular product, despite a stronger CD62L expression in EPL- and RAK-specific T cells as compared to HPV-specific T cells (S5 Fig).

Separate analysis showed that T cell memory phenotypes were extensively changed in the CD8+ but hardly in the CD4+ T cell subset (Fig 1E), while expression of the activation markers CD25, HLA-DR and CD38 was also largely limited to the CD8+ subset (Fig 1F). As far as is known (and disregarding the possibility of promiscuous HLA class II restriction [45]), peptides presented on HLA-DR to CD4+ T cells and which were used for stimulation (Table 1) were not restricted for any donor or patient HLA-ABC molecule. Consequently, EBV-specific CD4+ T cells may not have been stimulated by the EBV peptide pool, and therefore memory and activation markers on CD4+ T cells were not altered.

## Analysis of the TCR repertoire of the T cell product

Having demonstrated that stimulation of T cells with a pool of EBV peptides results in strong expansion of peptide-specific CD8+ effector and effector/effector-memory T cells, we next analyzed the T cell receptor β-chain (TCRβ) repertoire before and after peptide stimulation. To this end, we amplified the TCRβ of flow cytometry-sorted CD8+ T cells via high-throughput sequencing (HTS) (see S6 Fig for multimer sorting gating strategy and purity). For comparability, the same amount of DNA (100 ng per analysis)—representing the equivalent number of

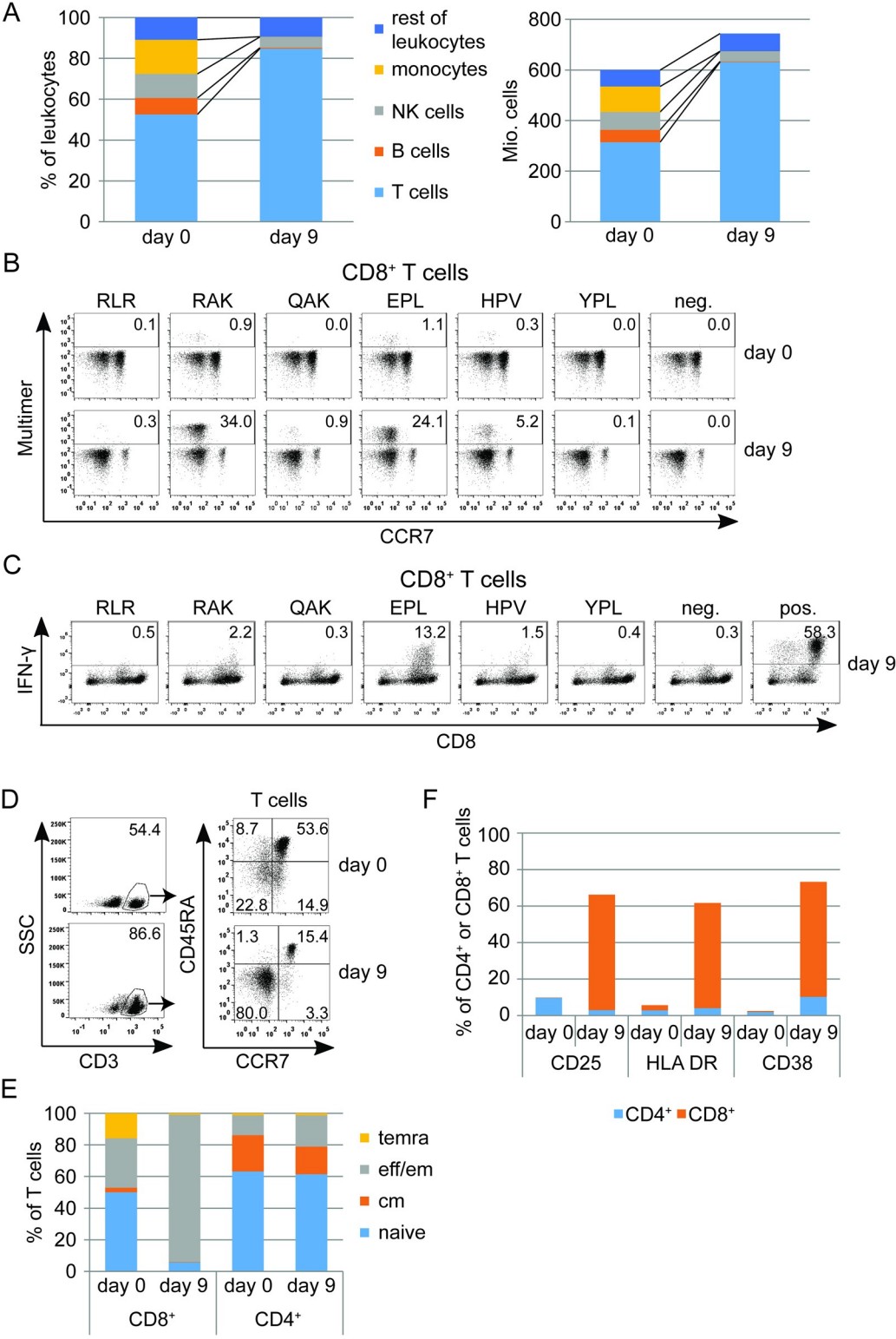

**Fig 1. Manufacture of EBV-specific T cells.** (A) Composition of the apheresis product that served as starting material (day 0) and the resulting T cell culture after stimulation with EBV peptides (day 9). Left panel: Proportions of different cell types in CD45$^+$ cells (monocytes: CD14$^+$SSCl$^{ow}$, NK cells: CD56$^+$, B cells: CD19$^+$, T cells: CD3$^+$, rest of leukocytes). Right panel: Composition in absolute cell numbers. (B) Percentage of peptide-specific T cells assessed by flow cytometry using peptide-MHC multimers (RLR, HLA-A*03:01; QAK, RAK, HLA-B*08:01; YPL, HPV, EPL, HLA-B*35:01) on day 0 and

day 9. (C) IFN-γ secretion of peptide-specific T cells after restimulation with single peptides on day 9, assessed by intracellular cytokine staining (neg.: unstimulated CD8$^+$ T cells, pos.: stimulation of T cells with ionomycin). (D) Flow cytometric analysis of T cell memory/differentiation markers on day 0 and day 9. Plots on the right side are pre-gated on CD3$^+$ cells. temra = terminally differentiated effector memory T cells (CCR7$^-$ CD45RA$^+$), eff/em = effector/effector memory T cells (CCR7$^-$CD45RA$^-$), cm = central memory T cells (CCR7$^+$ CD45RA$^-$), naïve = naïve T cells (CCR7$^+$CD45RA$^+$). (E) Percentage of T cell subsets within CD4$^+$ and CD8$^+$ T cells. (F) Flow cytometric analysis of T cell activation markers CD25, HLA-DR and CD38 within the CD4$^+$ and CD8$^+$ T cell compartment.

T cell rearrangements (approximately 14,500 T cells)—was employed for each library preparation. Within a first study, we were able to demonstrate that analyzing this constant amount of T cells reliably reflects T cell composition and T cell diversity [46].

Following this approach, we observed a strong change in the usage of TCR Vβ segments of the sorted CD8$^+$ T cells before and after EBV peptide stimulation (Fig 2A, left panel). While the proportion of Vβ19, 20, and 4 was reduced over 9 days of cultivation, we observed an expansion of Vβ6 and Vβ7 chains. Next, we enriched EBV epitope-specific CD8$^+$ T cells by flow cytometry cell sorting based on peptide-MHC multimer binding on day 9. Interestingly, individual patterns of Vβ usage were characteristic for each EBV epitope (Fig 2A, right panel), with predominant Vβ6 usage in EPL and HPV multimer-enriched CD8$^+$ T cells, and Vβ7 and Vβ4 predominant in RAK-enriched T cells.

Individual clonotypes were defined as TCRβ complementarity-determining regions 3 (CDR3) DNA sequences with a percentage of reads equal to or above the cut-off of 0.01%. We compared the frequencies of the 25 most abundant TCRβ clonotypes at day 0 and day 9 of the CD8$^+$ T cells, ordered by read frequency in descending order (Fig 2B, left panel). While the percentage of the most common clonotype (labeled by arrow) of the total CD8$^+$ T cell fraction on day 0 was 1.4%, the most dominant clonotype on day 9 reached 14.5%.

Analysis of the clonotype distribution of multimer-sorted T cells (Fig 2B, right panel) revealed a steep distribution curve for epitope HPV-sorted T cells with the most dominant single TCRβ clonotype (CASGTEAFF) representing 38.8% of all HPV-sorted TCRβ sequence reads. In contrast, RAK- and EPL-sorted CD8$^+$ T cells showed a less steep distribution curve, indicating a higher variety of different TCRβ clonotypes.

A higher percentage of the most common clonotype correlated with a lower total number of different clonotypes per sample. This correlation was also present in clonotype numbers in CD8$^+$ T cells before and after peptide stimulation (Fig 2C, left panel). At day 0, we identified 1,957 different TCRβ clonotypes (cutoff 0.01% of reads) derived from an equivalent of approximately 14,500 T cells (100 ng input DNA). This number was reduced to 471 clonotypes after stimulation, and 276 of these were shared in both samples. These 276 clonotypes accounted for 27.4% of total sequence reads on day 0 and for 80.8% of reads on day 9. Of these 276 clonotypes, 208 were found in multimer-sorted populations: 108 in EPL-, 97 in RAK-, and 96 in HPV peptide-MHC multimer-binding T cells (overlapped clonotypes had only minor frequencies). These 208 clonotypes accounted for 16.2% of all CD8$^+$ TCRβ sequence reads of the healthy donor (day 0). This fraction increased to 77.5% of all detected CD8$^+$ T cell TCRs in the peptide-stimulated T cell product (day 9).

After flow cytometric cell sorting with the three peptide-MHC multimers, a total of 327 clonotypes were present in cells sorted with EPL multimer, 341 clonotypes in RAK-sorted cells, and 313 clonotypes in HPV-sorted cells. Multimer sorting gate was kept stringent to achieve a sorting purity above 98% (S6 Fig). To clearly identify epitope-specific clonotypes and remove both overlapping and contaminant cells in multimer-sorted populations, we established two additional filters: (1) a frequency cutoff of 0.1% before and after multimer sorting, and (2) a requirement that epitope-specific clonotypes were at least ten times more highly enriched in

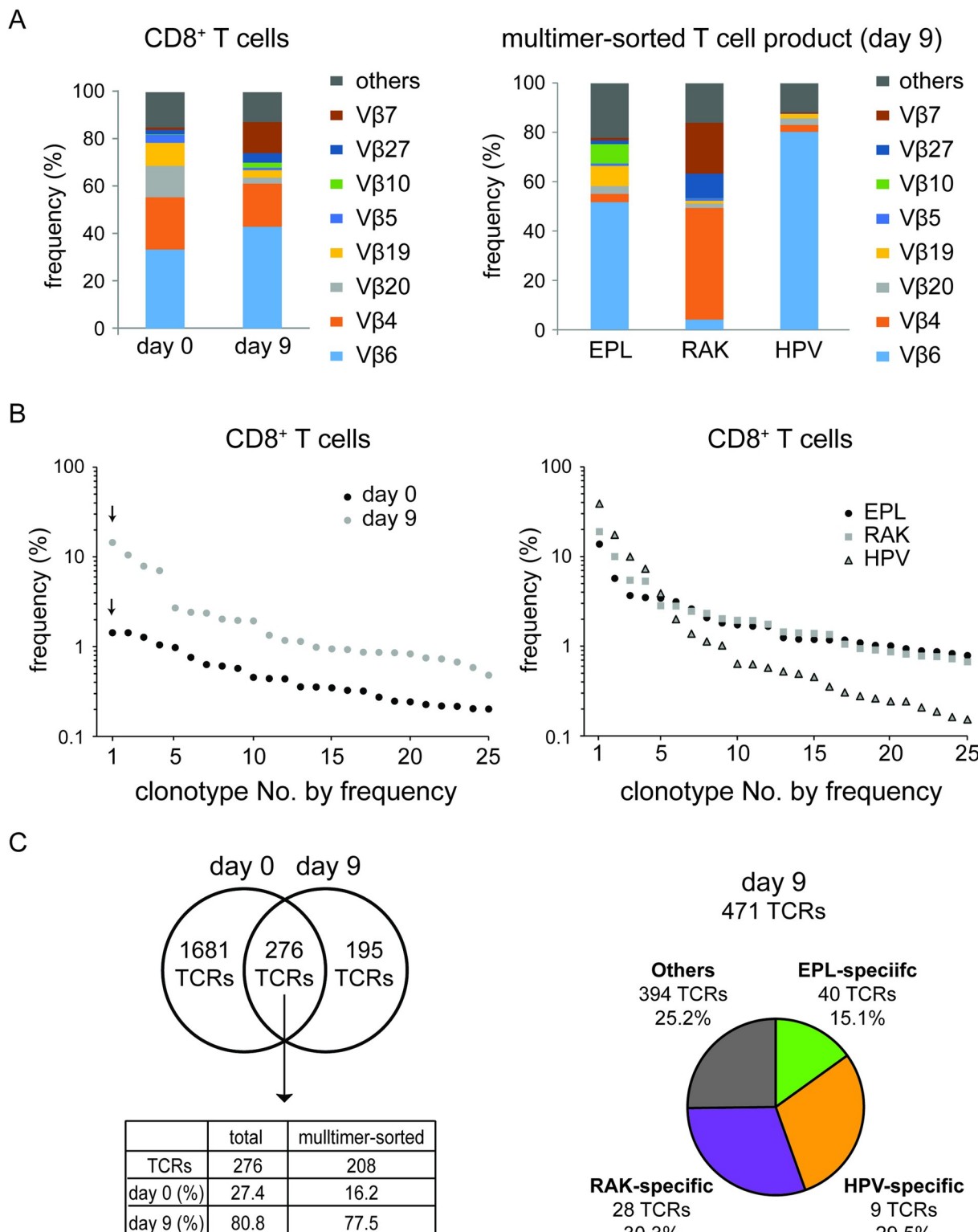

**Fig 2. Global and EBV-specific TCR repertoire of the T cell product.** (A) Vβ usage. Left panel: Percentage of Vβ subgroup usage within CD8+ T cells before (day 0) and after (day 9) peptide stimulation. Right panel: Vβ subgroup analysis of flow cytometry-sorted MHC multimer binding T cells (EPL, RAK, and HPV) on day 9 after peptide stimulation. (B) Individual clonotype distribution within CD8+ T cells on day 0 and day 9 (left panel) and in multimer-sorted T cells on day 9 (right panel). Each dot represents the frequency (percentage of all sequencing reads) of a single TCRβ clonotype. The 25 most frequent clonotypes of each sample are illustrated. (C) Left panel: Overlap of the number of

TCRβ clonotypes within CD8+ T cells on day 0 and day 9. The table shows the presence of the 276 shared TCRs from days 0 and 9 in total CD8+ T cells and in multimer-sorted CD8+ T cells from day 9, along with their cumulative percentage in total CD8+ T cells per day. Right panel: Number of epitope-specific TCRβ clonotypes and their proportion of cumulative TCRβ sequence reads within the overall CD8+ T cell product on day 9.

one of the multimer-sorted cultures than in the other two (ternary exclusion criterion). This analysis resulted in the identification of 40 EPL-, 28 RAK-, and 9 HPV-specific TCRs. (Fig 2C, right panel, epitope-specific clonotype identification in S7 Fig, TCR overview in S2–S4 Tables). Notably, the 77 epitope-specific clonotypes represented 74.7% of all TCRβ reads on day 9. This finding confirmed that day 9 peptide-expanded T cells were dominated by EBV epitope-specific CD8+ T cells. Among these, the 9 HPV-specific clonotypes accounted for 29.5% of all CD8+ clonotypes on day 9, which again reflects the steep distribution curve shown in Fig 2B (right panel) and the high proportion of dominant clonotypes such as CASGTEAFF in the HPV-sorted fraction.

To illustrate how EBV peptide-sorted T cell clonotypes expanded after peptide stimulation, the 30 most common TCRβ clonotypes on day 0 and day 9 as well as the 10 most common clonotypes of EPL-, RAK-, and HPV-sorted T cells are listed in Table 2. Therein, the most dominant TCRβ clonotype for 3 EBV peptides is color-coded according to the peptide. Peptide stimulation resulted in a strong expansion of the dominant clonotypes for the three peptides. The dominant HPV-sorted TCRβ clonotype (CASGTEAFF) was also the most dominant one in day 9 CD8+ T cells and had been expanded 24-fold as compared to day 0. Within the multimer-sorted T cell fraction, this specific clonotype accounted for 38.8% of sequencing reads. Similar results were obtained for EPL and RAK dominant clonotypes. However, the HPV-specific TCRβ clonotype CASGTEAFF was found three times due to different DNA sequences coding for the identical amino acid sequence. Overall, these three clonotypes accounted for 66.2% within the HPV-sorted fraction.

## T cell expansion after adoptive T cell transfer

To follow the *in vivo* fate of adoptively transferred peptide-stimulated T cells, we analyzed the peripheral blood of the patient before and after transfer. Fig 3A shows T cell immune reconstitution in absolute T cell numbers after allo-SCT. Between day 34 and day 89, we observed massive expansion of CD4+ cells that was caused by the relapse of the underlying CD4+ AITL, as confirmed on a molecular level by preponderance of a single T-cell clonotype in a lymph node (S1 Fig). This hematologic relapse was accompanied by high fever and EBV reactivation emerging on day 66 and peaking on day 89 with 140,000 copies per ml peripheral blood (Fig 3A). The patient received four Rituximab doses weekly, starting on day 68, and an unseparated donor lymphocyte infusion (DLI) containing 5.0 Mio. CD3+ T cells/kg body weight on day 76 (S1 Table). No further therapy was given at that point. Over the course of the following 21 days, T cell counts strongly decreased. We therefore decided to generate an EBV-derived peptide-stimulated T cell product from frozen DLI portions. This product was transferred at a dose of 1.0 Mio CD3+ T cells/kg body weight on day 105 post-allo-SCT. As shown in Fig 3A, CD8+ T cells expanded after adoptive transfer for 8 days (day 105 to 113), followed by a decline over 13 days until day 126 and stable maintenance thereafter.

When the TCRβ repertoire within the donor's PBMC fraction used as DLI (Fig 3B, day 0) was analyzed, we obtained 2375 clonotypes within the CD4+ compartment and 1957 clonotypes within the CD8+ compartment, which is a typical degree of TCR diversity observed in healthy donors with the assay used here [46]. Consistent with an expected narrowing of the TCR repertoire following allo-SCT [48], our patient's TCRβ repertoire was strongly reduced in

**Table 2. Expansion of distinct clonotypes after EBV-derived peptide stimulation.**

| CD8+ T cells day 0 | | CD8+ T cells day 9 | | HLA multimer-sorted T cell product | | |
|---|---|---|---|---|---|---|
| reads (%) | CDR3 | reads (%) | CDR3 | reads (%) | CDR3 | peptide |
| 1.426 | CASTTPGGRNEKLFF | 14.464 | CASGTEAFF | 13.732 | CASRDRVGSEAFF | EPL |
| 1.426 | CATSRARGSGANVLTF | 10.491 | CASSSQRQGRTYEQYF | 5.709 | CASSDSGTTFNEQFF | |
| 1.271 | CSAKGSLETEAFF | 7.862 | CASSTSRGAGNTIYF | 3.672 | CASSDSGIHNSPLHF | |
| 1.044 | CASSYPGQLNEKLFF | 7.020 | CASGTEAFF | 3.487 | CASSDTSALNTEAFF | |
| 0.975 | CASSQDPGNTEAFF | 2.700 | CASGTEAFF | 3.425 | CAISTGDSNQPQHF | |
| 0.757 | CASSEGYSNQPQHF | 2.407 | CASTSSRGGGNTIYF | 3.132 | CASRGGQGQETQYF | |
| 0.629 | CSASDTGISGANVLTF | 2.359 | CASGNEQYF | 2.604 | CASRTGEVNEQFF | |
| 0.605 | CASGTEAFF | 2.028 | CASSQASYVQGDGYTF | 2.081 | CASSTGDSNQPQHF | |
| 0.572 | CASSQDYAGHQPQHF | 1.959 | CASRDRVGSEAFF | 1.817 | CASGTFDSNQPQHF | |
| 0.454 | CSAKGGYDTEAFF | 1.929 | CASGSEAFF | 1.730 | CASSDSGMTEAFF | |
| 0.440 | CASSLNGEGTYEQYF | 1.339 | CAISTGDSNQPQHF | 38.810 | CASGTEAFF | HPV |
| 0.436 | CSVRGRENSPLHF | 1.170 | CASSPGGGTEAFF | 17.472 | CASGTEAFF | |
| 0.356 | CASSMALTATNEKLFF | 1.144 | CASSSLNTEAFF   P2 | 9.966 | CASGTEAFF | |
| 0.354 | CASSPTGNTEAFF | 0.984 | CSARDRGDTYEQYF | 7.303 | CASGSEAFF | |
| 0.347 | CASSTSRGAGNTIYF | 0.940 | CASRTGEVNEQFF | 3.890 | CASRPTGFDGYTF | |
| 0.326 | CASSQESDYGYTF | 0.925 | CSAGQGEGYEQYF | 1.998 | CASGNEQFF | |
| 0.319 | CASSQADSFSGNTIYF | 0.863 | CASRPPGPFYEQYF | 1.383 | CSAALRPVPRTGYTF | |
| 0.273 | CASSQESGHLNTEAFF | 0.862 | CASSTGDVNQPQHF | 1.130 | CASSSRSGELFF | |
| 0.246 | CASSAETGGGEKAFF | 0.853 | CASSQGLPLNTEAFF | 1.019 | CASIPRTKTEAFF | |
| 0.242 | CASRDRVGSEAFF | 0.827 | CASSYGPYEQYF | 0.636 | CASGNEQFF | |
| 0.226 | CASSQGPNYEQYF | 0.746 | CASSDSGIHNSPLHF | 18.947 | CASSTSRGAGNTIYF | RAK |
| 0.217 | CASSIGQAYEQYF | 0.729 | CASRGGQGQETQYF | 9.985 | CASSSQRQGRTYEQYF | |
| 0.216 | CASSESPAGEQYF | 0.672 | CASSDSGTTFNEQFF | 5.449 | CASSQGLPLNTEAFF | |
| 0.203 | CSARDPGSSYEQYF | 0.583 | CASSSLNTEAFF   P2 | 5.294 | CASTSSRGGGNTIYF | |
| 0.202 | CASSLAPGYLYYEQYF | 0.479 | CSARGASPQANYGYTF | 2.834 | CSAGQGEGYEQYF | |
| 0.195 | CSARGGETEAFF | 0.432 | CASSDTSALNTEAFF | 2.810 | CASSSLNTEAFF   P2 | |
| 0.192 | CASSEAGTGRSEQYF | 0.427 | CASSYSSFRGGNSPLHF | 2.442 | CASSSLNTEAFF     P2 | |
| 0.189 | CASSKTMGMGTDTQYF | 0.414 | CASGNEQFF | 2.330 | CASSLIASGGYNEQFF | |
| 0.186 | CASGTEAFF | 0.403 | CASSSLNTEAFF | 2.025 | CASSQGVTDYWNEQFF | |
| 0.175 | CASSLSYEQYF | 0.394 | CASSQPGGLEQYF | 1.946 | CASSQGTGFNYGYTF | |

Ranking of the top 30 TCRβ clonotypes in donor-derived CD8+ T cells before peptide stimulation (day 0, left column), after peptide stimulation (day 9, middle column), and after peptide stimulation with HLA multimer FACS sorting on day 9 (right column, top 10 clonotypes for each specificity). Clonotypes with identical CDR3 peptide sequence are presented separately in case of a different underlying CDR3 DNA sequence and marked in red. The most dominant clonotype per multimer-sorted T cells are color-coded regarding the peptide used for sorting. P2: public clonotypes previously published. [47]

both compartments (CD4+: 236, CD8+: 108 clonotypes) on day 60 after allo-SCT (before DLI and adoptive transfer of EBV peptide-stimulated T cells). In line with hematologic relapse, one clone was predominant in the CD4+ fraction of peripheral blood (CSARDRTGSEKLFF). This clone represented the CD4+ AITL, which was confirmed by analysis of DNA retrieved from a lymph node biopsy at the time of initial diagnosis (S1 Fig). On day 120 (fifteen days after adoptive transfer of peptide-stimulated T cells), we observed an increase in T cell diversity, which was higher in CD8+ T cells (645 clonotypes) than in CD4+ T cells (402 clonotypes). This suggested that adoptive transfer on day 105 contributed to diversification of the patient's TCR repertoire, in particular through transfer of EBV-specific CD8+ T cells.

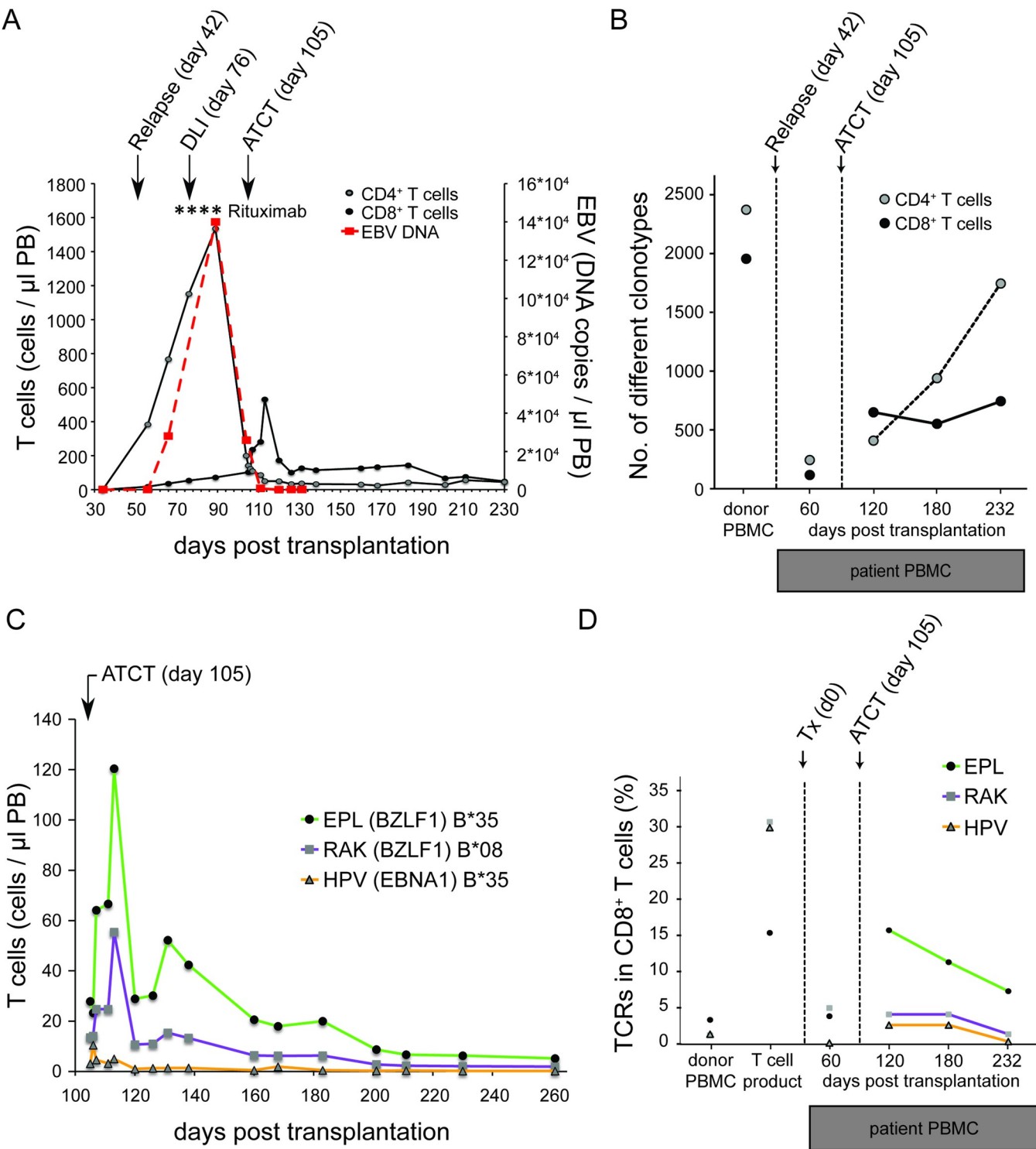

**Fig 3. T cell and clonotype expansion after allogeneic stem cell transplantation and adoptive transfer of EBV-specific T cells.** (A) Flow cytometric monitoring of absolute numbers of CD4+ and CD8+ T cells and EBV DNA copy number in peripheral blood. Relapse of the CD4+ T cell lymphoma was detected on day 56 in peripheral blood. Time points of Rituximab application are marked with an asterisk (*). ATCT = adoptive T cell transfer. (B) TCR clonotype diversity in CD4+ and CD8+ T cells in peripheral blood of the patient. For comparison, the diversity in donor's PBMC is shown. (C) Flow cytometric monitoring of peripheral blood CD8+ T cells using HLA peptide-MHC multimers (EPL, RAK, HPV) on the day of ATCT (day 105) and thereafter. (D) Cumulative frequency of TCR clonotypes specific for each of the epitopes EPL, RAK, and HPV in CD8+ T cell populations. Data points for donor's PBMC, T cell product after peptide stimulation, and peripheral blood of the patient before (day 60) and after ATCT (day 120, day 180, and day 230) are shown.

Multimer staining was used to track EBV-specific T cells until day 232 after allo-SCT (Fig 3C). We observed a strong expansion of EPL- and RAK-specific CD8+ T cells, which were two of the three dominant EBV specificities in the T cell product. On day 105, immediately before adoptive transfer, a small fraction of HLA multimer-binding CD8+ T cells had been detectable in peripheral blood. Over the course of 8 days after transfer, RAK- and EPL-multimer-binding CD8+ T cells strongly expanded *in vivo* (from 14/μl to 55/μl and from 28/μl to 120/μl, respectively). Analysis of the TCRβ repertoire (Fig 3D) before adoptive T cell transfer (ATCT) on day 60 revealed that the patient had not mounted a significant T cell response against EPL, RAK, and HPV epitopes: Only 6 of 77 epitope-specific clonotypes were detectable. This situation had changed 15 days after adoptive transfer of peptide-stimulated T cells (day 120), when the total number of different multimer-binding clonotypes present had increased to 61. In comparison to day 60, by day 120 EPL- and HPV-specific T cell read frequencies increased significantly (EPL: from 3.67% to 15.43%, HPV: 0.00% to 2.37%), while RAK was relatively stable (RAK: 4.78% to 5.06%), Thereafter, clonotype diversity in CD8+ T cells remained rather constant (Fig 3B), while epitope-specific clonotypes gradually declined until day 232 but were detectable throughout the observation period (follow-up of epitope-specific TCRs in S5 Table).

A complete frequency analysis of the 77 EBV epitope-specific clonotypes is shown in Fig 4A as a heat map (comparison in the T cell product before and after multimer sort in S8 Fig). At the beginning of the EBV peptide-stimulated T cell product manufacture (which at this point represents the donor's natural T cell repertoire on day 0), we found 55 EBV-specific clonotypes to be present in the patient at this time. These clonotypes represent 5.43% of the CD8+ TCRβ repertoire before peptide-stimulation on day 0 (S5 Table). 15 days after adoptive transfer (day 120), the 77 EBV epitope-specific clonotypes accounted for 22.86% of all TCR gene reads found in the patient. On our last measurement (day 232), 45 EBV-specific clonotypes remained detectable, representing 8.49% of all TCR reads. Thus, by adoptive transfer of peptide-stimulated T cells, we reinstalled a large part of the donor's specific T cell repertoire targeting three EBV epitopes, which is especially reflected by the dominant clonotypes for each epitope (Fig 4B upper panel and Table 2). EBV-specific T cell reconstitution in the patient included two previously described (and thus public) clonotypes specific for EPL and RAK (Fig 4B, lower panel, and Table 2), but one of these became undetectable on day 232 [31,33]. It is noteworthy that neither dominant nor proven public clonotypes were found among the 6 EBV-specific clonotypes on day 60 in the patient, when EBV began to reactivate in the period before ATCT.

## Discussion

Adoptive transfer of EBV-specific T cells for the treatment of EBV-associated lymphoma in the immunocompromised host has been shown to effectively mediate virus control [50]. Furthermore, it has been demonstrated that adoptively transferred EBV-specific T cells contribute to long-term immunity [20]. However, although several dominant EBV-derived T cell epitopes and their HLA restriction were identified over the past decades [30–43], it remains unclear which and how many TCRs recognize those epitopes and are being expanded *in vivo*. Follow-up in patients after primary infection with EBV suggests few clonotypes with high frequencies dominate epitope-specific responses long-term [51]. Similar observations were made after adoptive transfer of EBV-specific T cells [52,53], thus pointing towards TCR clonotypes of potential clinical interest. Beyond single clonotypes, EBV-specific T cell frequencies, repertoire diversity, and long-term survival of TCR clonotypes contribute to control active EBV infection [54], latency [55], and EBV-associated malignancies [56–58].

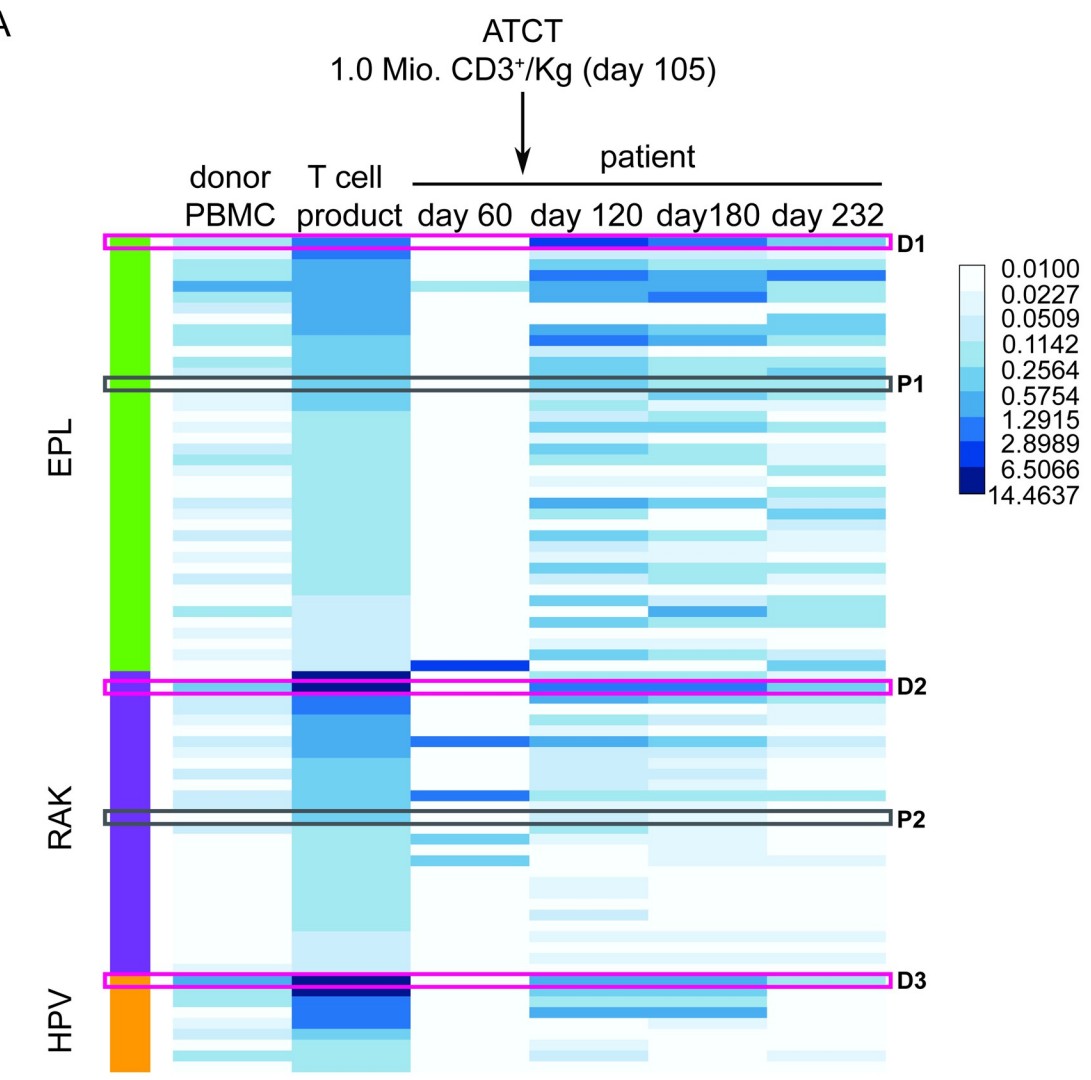

**Fig 4. Frequencies of specific TCRβ clonotypes before and after adoptive T cell transfer.** (A) Frequency heatmap of individual epitope-specific TCR clonotypes in donor PBMC, T cell product, and four time points after transplantation in the patient (day 60, day 120, day 180, and day 232). Clonotype frequency is displayed as a percentage from 0.01% (limit of detection) to 14.4637 by increasing colour depth. Each row represents one specific TCR clonotype. Identified public TCR sequences (P1 and P2) previously published [47,49] are shown in grey boxes. The most dominant clones (D1-3) within each specificity are shown in pink boxes. ATCT: adoptive T cell transfer. (B) Frequencies of public clonotypes (P) and dominant clonotypes (D) in different samples.

**B**

**dominant clonotypes (D)**

| D# | donor PBMC | T cell product | sort day 9 | patient | | | | CDR3 sequence | epitope |
|---|---|---|---|---|---|---|---|---|---|
| | | | | day 60 | day 120 | day 180 | day 232 | | |
| D1 | 0.242 | 1.959 | 13.732 | 0.000 | 3.474 | 2.499 | 0.482 | CASRDRVGSEAFF | EPL |
| D2 | 0.347 | 7.862 | 18.947 | 0.000 | 1.969 | 2.169 | 0.532 | CASSTSRGAGNTIYF | RAK |
| D3 | 0.605 | 14.464 | 38.810 | 0.000 | 1.011 | 0.946 | 0.135 | CASGTEAFF | HPV |

**public clonotypes (P)**

| P# | donor PBMC | T cell product | sort day 9 | patient | | | | CDR3 sequence | epitope |
|---|---|---|---|---|---|---|---|---|---|
| | | | | day 60 | day 120 | day 180 | day 232 | | |
| P1 | 0.019 | 0.338 | 2.081 | 0.000 | 0.492 | 0.219 | 0.154 | CASSTGDSNQPQHF | EPL |
| P2 | 0.016 | 0.261 | 1.053 | 0.000 | 0.058 | 0.031 | 0.000 | CASSSLNTEAFF | RAK |

In this study, we manufactured and extensively analyzed at a molecular level the fate of adoptively transferred T cells enriched for specificity against multiple EBV-derived epitopes. *In vitro* stimulation with defined EBV latent and lytic epitopes resulted in a strong expansion of three EBV epitope-specific CD8$^+$ T cell populations (two lytic antigen epitopes and one latent antigen epitope). Each of the three specificities were reconstituted *in vivo* and T cells were maintained in the patient for at least five months. Furthermore, we comprehensively characterized the T cell clonotype repertoire for each of the epitope specificities by combining flow cytometry and high-throughput sequencing of TCRβ rearrangements.

This study was carried out in an EBV-seropositive patient who, after chemotherapy and allo-SCT for a CD4$^+$ AITL, suffered from relapse and simultaneous increase of peripheral blood EBV load, most likely associated with the leukemic relapse of the lymphoma. EBV-encoded RNA (EBER) *in situ* hybridization (IsH) analysis of AITL in the lymph node at diagnosis revealed the AITL tumor to be EBV negative (S2 Fig). This suggests that high EBV titers derived from an active infection site other than the AITL. However, simultaneous high EBV viremia and AITL relapse suggests a strong association between AITL and EBV infection, including a possible role in its pathogenesis [2] and increased aggressiveness [4].

Despite DLI and Rituximab treatment, adoptive transfer of T cells enriched for EBV epitopes was provided due to persistence of severe B-symptoms and increasing levels of EBV viral DNA in the peripheral blood. In addition, on day 60 post allo-SCT, we observed a complete lack of EBV peptide-MHC multimer-binding T cells for the six known HLA-relevant epitopes. Owing to the risk of recurrent EBV reactivation [59] and its concomitant occurrence with AITL relapse, we believe adoptive transfer of EBV-specific T cell products hold promise to support the treatment of AITL by controlling EBV infection [60,61].

We cannot exclude at this point that the conventional DLI may indeed have contributed to the surviving EBV memory T cell pool; nonetheless, we were able to track EBV-specific T cell clonotypes coming from the adoptive transfer that persisted long-term in the patient. In contrast to the conventional DLI, EBV-specific T cells immediately expanded after adoptive transfer, resulting in a profound cytokine release syndrome (S4 Fig).

Variance between *ex vivo* and *in vivo* expansion of epitope-specific cells, especially against antigens of latency and lytic phases, may arise from differences in differentiation phenotype and epitope availability at both stages. Although T cells in the cellular product mostly lacked CCR7 expression (CCR7$^-$CD45RA$^-$), both lytic BZLF-1 EPL- and RAK-specific T cells had stronger CD62L expression than latent EBNA1 HPV-specific T cells (S5 Fig). CD62L expression facilitates homing into lymph nodes through adhesion to high endothelial venules (HEV) [62], with a closer association to central memory phenotype (CD62L$^-$CD45RA$^-$) [63]. This difference may indicate a higher proliferative strength, a stronger expansion (as seen in Fig 3C), less exhaustion, and better homing ability of the lytic antigen-specific T cells. An alternative hypothesis to the difference in expansion between lytic and latent antigen-specific T cells is the availability of their target epitope: The strong expansion of EPL- and RAK-specific T cells in peripheral blood and cytokine release (S4 Fig) happened directly after transfer, when ongoing EBV viremia (suggestive for lytic cycle) may have provided profound lytic epitope presentation. EBV viremia decrease over time and lesser epitope availability would explain why few clonotypes remained and survived long-term, which is expected as a natural modulation of the immune T cell response. On the other hand, HPV-specific T cells (latent antigen EBNA1) strongly expanded *ex vivo* but not in peripheral blood *in vivo*, either due to its homing to an active infection site, such as a lymph node, or the unavailability of their target antigen. In either case, we demonstrated that *in vitro* T cell stimulation with a defined set of peptides results in broad spectrum TCRβ repertoire expansion of various dominant clonotypes for relevant latent and lytic epitope specificities, which then get further selected in *vivo* based on need during the

course of viral reactivation. They might therefore contribute to reinstallation of a natural occurring immunity similar to the one observed in the donor.

Approximately 10 days after adoptive transfer of EBV-specific T cells and approximately 40 days after conventional DLI, the patient developed a skin rash, high bilirubin and liver enzyme elevation in the sense of acute GvHD (see S9 Fig). We cannot clearly attribute this to either DLI or the T cell product. The conventional DLI contained approximately 53.6% naïve T cells (equivalent to day 0 analysis for the manufacturing, see Fig 1D). In total 201 Mio. naïve T cells were infused with the conventional DLI (5.0 Mio. CD3$^+$ T cells/kg). In contrast, a total of only 11.55 Mio. naïve T cells were infused along with the EBV-specific T cell product (1 Mio. CD3$^+$ T cells/kg). We would speculate retrospectively that GvHD was mainly due to the conventional DLI, and that the cytokine release syndrome induced by the EBV-specific T cell product augmented the naïve T cell response. The patient received 1mg/kg Prednisone and responded very quickly to this treatment, which is also reflected in the rapid decline of CD8$^+$ T cells in the peripheral blood. However, despite the high dose of steroids, the patient did not reactivate EBV again and EBV-specific T cells persisted at a lower level demonstrating steroid resistance.

Our TCR high-throughput sequencing approach is based on samples of 100 ng cellular DNA and therefore has a limited resolution of 14,500 T cells, with a frequency cut-off of 0.01% (approximately 100 reads per T cell). This is suitable to obtain insights into clonotype diversity but will not detect every virus-specific TCR clonotype in patient samples or T cell products. High-throughput sequencing was able to reveal multiple EBV-specific clonotypes even in the complex T cell repertoire of the healthy donor, whose cells were used to manufacture EBV-specific T cells for adoptive transfer.

Expansion of EPL-, RAK-, and HPV-MHC multimer-binding T cells after peptide stimulation correlated with the expansion of distinct clonotypes, as shown by flow cytometry and TCRβ high throughput sequencing. However, we found TCRs in the MHC multimer-sorted CD8$^+$ population that were not enriched (as compared with unsorted populations) but were still detectable, presumably due to unspecific MHC multimer-binding. Nonetheless, this fraction represented the purest pool of T cells for a defined specificity (sorting purity above 98%) and was used to for multimer ternary exclusion.

Of note, the use of peptide-MHC multimer binding introduces a bias for TCRs with higher affinities because the affinity threshold required for multimer binding is higher than the one for T cell activation [64]. Therefore, we would conclude both BZLF1- and EBNA1-specific clonotypes identified in this study to have relatively higher affinity levels.

We mapped TCR sequences from peptide-MHC multimer-binding T cells back to the unsorted T cell pools before (d0) and after (d9) peptide stimulation. Using this strategy, we were able to identify previously described TCRβ sequences [31,32] and numerous new and naturally occurring clonotypes with relatively high frequencies in the normal donor. For example, three sequences with three distinct specificities found in the donor (Table 2, highlighted) accounted for 1.1% of the donor's repertoire. These were among the donor's 30 clonotypes with the highest frequencies and persisted long term after adoptive transfer in the patient. The presence of dominant clonotypes for single peptide specificities is reinforced by the fact that three different DNA sequences coded for HPV-specific TCRβ clonotype CASGTEAFF.

TCR clonotypes complying with both inclusion criteria (frequency above 0.1% before and after multimer sort in T cell product and multimer ternary analysis) were identified as EBV epitope-specific. The frequency cutoff of 0.1% was selected to reduce noise, while ternary analysis allows us to exclude unspecific multimer-binding clonotypes. Using both criteria combined, we were able to identify EBV epitope-specific T cells which clearly dominate the T cell

product (77 of 471 clonotypes account for 74.8% of reads) and persist long-term after adoptive transfer.

It is relevant that such few EBV-specific clonotypes were detectable on day 60, while the patient's EBV viral load in peripheral blood was increasing. In contrast, EBV epitope-specific clonotypes from the unsorted and MHC multimer-sorted CD8[+] populations on day 9 were found 15 days after adoptive transfer (day 120) with a significant increase in frequencies and diversity of TCR clonotypes (Fig 4A). Therefore, we could see an association between the presence of several EBV epitope-specific clonotypes on day 120 and the absence of EBV DNA in the peripheral blood thereafter. Due to the association of EBV infection and AITL relapse, EBV control could have positively influenced AITL regression, as has been observed in PTLD [65].

Success of ATCT after allo-SCT depends on restoring immunity against viruses without viral reactivation, in the absence of Graft versus Host Disease (GvHD) [66]. Several indicators of restored EBV-specific T cell immunity are: (i) the persistence of adoptively transferred, functional virus-specific T cells [20,67–69], (ii) absence and regression of EBV-associated lymphomas [65,70–72], and (iii) control of virus reactivation and viremia *in vivo* [69,73–76]. We would therefore argue that the presence of EBV epitope-specific, expanded clonotypes in the T cell product, their long-term persistence in the patient, and lack of further EBV reactivation or relapse point to an important role of these clonotypes in controlling EBV, AITL, and other EBV-associated malignancies.

In conclusion, we were able to confirm the long-term presence of expanded, EBV epitope-specific CD8[+] T cell clonotypes following adoptive transfer in the patient, thereby restoring anti-EBV T cell immunity. To further validate these findings, a recently closed multicenter phase I/IIa clinical study (NCT02227641, EudraCT: 2012-004240-30) used this manufacture technique to generate T cell products with double specificity against CMV and EBV for patients after allo-SCT. The results of this study will further increase our knowledge on potentially protective virus-specific TCR repertoires after allo-SCT.

## Material and methods

### Ethics statement

The patient gave written informed consent prior to transplantation for extended immunomonitoring using standard flow cytometry, multimer analysis, and TCR HTS. The ethics committee of Friedrich-Alexander-University Erlangen-Nürnberg gave approval for this study (approval No.: 4388). In addition, the patient gave written consent for the attempt to cure using donor-derived EBV-specific T cells.

### EBV viral load analysis

EBV viral load measurement was carried out with whole blood EDTA as part of regular follow-up in the hospital for all allo-SCT patients. The QiaSymphony DSP Virus/Pathogen Mini Kit (QIAGEN, Hilden, Germany) was used for viral nucleic acid purification, while real-time PCR was established in-house and adapted from literature [77].

### *in situ* hybridization

EBER *in situ* hybridization was carried out on an AITL lymph node sample at time of diagnosis.

## Generation of EBV-specific T cells

EBV–specific peptides were generated in a GMP-conform fashion as described previously [68]. Peptides used for stimulation are shown in Table 1. In brief: frozen donor lymphocytes were obtained and thawed for Ficoll density centrifugation, yielding 826 x 10$^6$ PBMC. PBMC were frozen until use. 600 million PBMC were incubated with peptide mix for 2h. After subsequent washing steps, cells were incubated in a closed bag system for 9 days. Medium was added according to the manufacturing protocol on day 5. Quality assessment of the product included bacterial culture and eubacterial PCR, flow cytometric analysis, and trypan blue method for viability.

## Flow cytometry analysis of cultivated cells and peripheral blood

To quantify cell types, peripheral blood (50 µl per sample) was stained in TruCount tubes containing fluorescent beads (BD Biosciences) with the following antibodies: anti-CD8 FITC (clone SK1), anti-CD25 PE (clone 2A3), anti-CD14 PerCP (clone MφP9), anti-CD56 APC (clone B159), anti-CD19 PE-Cy7 (clone SJ25C1), anti-CD4 APC-Cy7 (clone RPA-T4), anti-CD3 V450 (clone UCHT1), and anti-CD45 V500 (clone HI30, all clones from BD Bioscience). After incubation at room temperature for 15 min, 450 µl of red cell lysis buffer (BD Biosciences) was added and samples were incubated for further 20 min. Cells were analyzed subsequently after staining using a FACS Canto II flow cytometer (Becton Dickinson). Leukocytes were gated as CD45$^+$ and lymphocytes as CD45$^{high}$CD14$^-$ cells. Within the lymphocyte population, T cells were determined as CD3$^+$, B cells as CD19$^+$, NK cells as CD56$^+$ cell populations. T cell subpopulations were analyzed for CD4 and CD8 expression. Cell counts/µl were calculated based on bead count and sample volume in TruCount tubes (BD Bioscience). Cultivated cells were stained with the same panel but without cell quantification by TruCount tubes.

For analysis of T cells with multimer staining, 1x10$^6$ cells either PBMC isolated from peripheral blood by Ficoll density centrifugation or taken from cultivated cells on day 0 and day 9, were stained with HLA-matched peptide-MHC pentamers (ProImmune, Oxford, UK), and subsequently counterstained with PE-fluorotag (Proimmune), anti-CCR7 FITC (clone 150503, R&D Systems, Minneapolis, MN, USA), anti-CD8 PerCP (clone SK1), anti-CD62L APC (clone DREG-56), anti-CD45RA PE-Cy7 (clone HI100), anti-CD4 APC-Cy7 (clone RPA-T4), and anti-CD3 V450 (clone UCHT1, all clones BD Biosciences). Cells were analyzed using a FACS Canto II flow cytometry analyzer (Becton Dickinson). Vital lymphocytes were gated in FSC vs. SSC. T cells were identified by their CD3 expression. T cell subpopulations were identified by CD4 and CD8 expression. T cells binding an EBV peptide-MHC multimer were analyzed within the CD8$^+$ T cell population.

Cultivated cells after harvest were further analyzed for IFN-γ production upon antigen-specific restimulation. Therefore, day 9 cells were restimulated with the epitopes RLR, RAK, QAK, EPL, HPV, or YPL (each peptide 0.5µg/ml), or PMA-ionomycin for positive control. To inhibit IFN-γ secretion, GolgiStop (BD Biosciences) was added for the time of restimulation (5 hours). Afterwards, cells were harvested and stained with the following surface markers: anti-CD3 PerCP (clone SK7), anti-CD8 PE-Cy7 (SK1), and anti-CD4 APC-Cy7 (clone RPA-T4). Then, cells were washed and treated with 250µl CellFix /Perm buffer (BD Biosciences) for 20 minutes, 4˚C. Cells were then washed with Perm-/Wash buffer (BD Biosciences) and subsequently intracellularly stained with anti-IFN-γ-FITC (clone B27) for 30 minutes at 4˚C. Afterwards, cells were once washed with Perm-/Wash-buffer and once with PBS. After cell fixation, samples could be analyzed by flow cytometry.

## Cell sorting

Whole blood samples (EDTA) were processed by density gradient centrifugation (Ficoll) to obtain mononuclear blood cells (PBMC). For flow cytometry sorting, PBMC were stained with anti-CD4 FITC (clone SK3), anti-CD8 PE (clone SK1), anti-CD14 PerCP (clone MφP9, all clones BD Biosciences, Franklin Lakes, NJ, USA), and anti-TCRαβ (clone BW242/412, Miltenyi Biotec, Bergisch Gladbach, Germany). Cells were gated on (i) vital lymphocytes in forward/side scatter, (ii) exclusion of doublets, and (iii) TCRαβ$^+$ CD14$^-$ T cells. Within the T cell population, CD4$^+$ and CD8$^+$ T cells were sorted into separate tubes (MoFlow, Beckman Coulter, Brea, CA, USA). A purity of > 98.0% was achieved as monitored by reanalysis of the sorted samples.

Multimer cell sorting was performed using HLA-matched peptide-MHC pentamers obtained from ProImmune. Of the EBV-specific expanded T cells (day 9), 40x10$^6$cells were incubated with RAK-HLA-B$^*$08:01-, 40x10$^6$cells with HPV-HLA$^*$B35:01-, and 18x10$^6$cells with EPL-HLA$^*$B35:01-multimers, according to manufacturer's recommendation. Afterwards cells were washed and stained with PE-fluorotag (ProImmune) binding to the peptide loaded HLA multimers, anti-CD8 FITC (clone SK1, BD Biosciences), anti-CD14 PerCP (clone MφP9, BD Biosciences), and anti-TCRαβ (clone BW242/412, Miltenyi Biotec, Bergisch Gladbach, Germany). Cells were gated on (i) vital lymphocytes in forward/side scatter, (ii) exclusion of doublets, and (iii) TCRαβ$^+$ CD14$^-$ T cells. Then, the CD8$^+$ multimer-binding population was sorted out and used for further analysis by TCRβ sequencing.

## DNA isolation

DNA was extracted from flow cytometry-sorted T cells using the Qiagen AllPrep DNA/RNA Mini Kit (Qiagen) according to the manufacturer's instructions. Quantification of the extracted DNA was done employing a Qubit 1.0 Fluorometer (Invitrogen, Carlsbad, CA, USA).

## Capillary electrophoresis

CDR3 length repertoires of TCRβ sequences were generated by using the BIOMED-2 primer sets for PCR-based clonality analysis [78]. The fluorescence-labeled amplicons were size-separated and detected via automated laser scanning by a 3130 Genetic Analyzer (Applied Biosystems; Darmstadt, Germany).

## High-throughput sequencing of TCRβ gene clonotypes

Amplification of TCRβ from 100 ng of cellular DNA (approximately 14,500 T cells) with multiplex PCR, sequencing of amplified TCRβ gene libraries (HiSeq2000), and data processing were performed as previously described [46]. Employing a two-step PCR strategy, the TCRβ amplicons were tagged with universal Illumina adapter sequences, including an additional barcode during a second amplification step, allowing parallel sequencing of several samples on Illumina HiSeq2000 (Illumina, San Diego, CA). Our amplicon sequences covered the entire CDR3 length and Vβ and Jβ segments in parts and using the Illumina paired-end technology (2x 100 bp) provided a high sequence accuracy.

The multiplex primers used contain a universal adapter sequence as a tail at the 5' end complementary to the 3' ends of second amplification adaptor primers. The adaptor PCR primers contained universal sequences that permitted solid-phase PCR on the Illumina Genome Analyzer (HiSeq 2000 Sequencing System). Primary amplification (final volume: 50 μL) was processed, including 100 ng DNA, 1.0 μM equimolar Vβ and Jβ primer pools, PCR buffer, 3mM

MgCl2, 0.2mM of each dNTP and 1U AmpliTaq Gold DNA Polymerase (Applied Biosystems, Foster City, CA). The amplification was performed on a DNA thermal cycler (GeneAmp1 PCR System 9700, Applied Biosystems) for 34 cycles at 62°C annealing temperature. All PCR products were purified using the QIAquick PCR Purification Kit (Qiagen) and diluted (final amount: 500 pg) for further amplifications. Adapter PCRs were set up with Phusion HF Buffer, 1.0 μM forward and reverse adapter primers, 0.05mM of each dNTP and 1U Phusion High-Fidelity DNA Polymerase (Finnzymes, Espoo, Finland). Secondary amplification was performed for 12 cycles at 58°C annealing temperature. Products were isolated from a 2% agarose gel using the Wizard1 SV Gel and PCR Clean-Up System (Promega, Mannheim, Germany). DNA concentration was determined via the Qubit1 1.0 Fluorometer (Invitrogen) [46]. Clonotypes were defined as TCRβ clonotypes with a percentage of reads equal to or above a 0.01% cut-off. Reads with frameshift or stop codon were considered as non-functional TCRβ rearrangements and excluded from analysis.

### Peptides

The manufacturing process involved stimulation of peripheral mononuclear cells by a fixed pool of peptides derived from various EBV proteins. The sequence of selected peptides, their HLA restriction, and reference is shown in Table 1. Peptides were synthesized by JPT Peptides Berlin (Germany) at a purity of 95%. All raw materials used for peptide synthesis were CE certified and all materials were fully synthetic. 5% contamination of the peptide product is considered to be smaller oligomers of the original design, due to inefficient elongation.

### Identification of EBV epitope-specific TCR clonotypes

We applied two criteria to TCR clonotypes found in the T cell product, either before or after multimer sort, to consider them as epitope-specific: First, we used a cutoff of 0.1%, representing approximately 10 T cells, to reduce noise. Then, we applied a ternary exclusion criterion based on the multimer enrichment ratio, defined as frequency after multimer sort / frequency before multimer sort. The enrichment factor for a given multimer must be at least ten times bigger than for the other two multimers to be considered epitope-specific.

### Supporting information

**S1 Fig. Lymphoma diagnosis and relapse.** (A) GeneScan analysis of T cell receptor gamma (TRG) and beta (TRB) demonstrated clonal T cell populations in the lymph node biopsy (day 166 before transplant). (B) HTS of TCRβ rearrangements of the lymph node permit identification of the lymphoma-specific TCRβ sequence (in bold). (C) The lymphoma-specific TCRβ sequence could be identified again in the recipient on day 60 after transplantation in peripheral blood.
(TIF)

**S2 Fig. No presence of EBV in the AITL.** (A) EBER *in situ* hybridization (IsH) of the lymph node at diagnosis. An infectious mononucleosis sample was used as positive control. (B) Staining of the lymph node at diagnosis. H&E: Hematoxylin and eosin.
(TIF)

**S3 Fig. Timeline of patient clinical history.** dpt: days post transplantation; ECOG: Eastern Cooperative Oncology Group; R-ICE: Rituximab, Ifosfamide, Carboplatin, and Etoposide Phosphate; GvHD: Graft-versus-Host Disease.
(TIF)

**S4 Fig. Serum cytokine levels after adoptive transfer of EBV-specific T cells (ATCT)**
(TIF)

**S5 Fig. T cell differentiation markers in the cellular product.** Multimer binding T cells were gated on CD8+ T cells. Plots were gated on multimer-binding T cells. Numbers indicate percentages.
(TIF)

**S6 Fig. Sorting of peptide-MHC multimer-binding T cells.** (A) Flow cytometric analysis gating for peptide-MHC multimer sorting. (B) Re-analysis of peptide-MHC multimer-sorted cells. Numbers indicate percentages.
(TIF)

**S7 Fig. Identification of epitope-specific T cells.** Scatter plots show frequency before and after multimer sort of the T cell product on day 9 of T cell clonotypes with a frequency above 0.1% in both populations. Each dot represents a single TCR clonotype. Red dots symbolize TCR clonotypes that pass a ternary exclusion criterion of at least ten times the multimer enrichment ratio for one multimer as compared with the other two and were, therefore, identified as epitope-specific.
(TIF)

**S8 Fig. Enrichment of epitope-specific clonotypes after sorting on day 9.** Frequencies of epitope-specific clonotypes (77 TCRs) is shown in unsorted CD8+ T cell product and MHC multimer-sorted sample. Clonotype frequency is displayed as a percentage from 0.01 (limit of detection) to 14.4637 by increasing colour depth. Each row represents one specific TCR rearrangement. Identified public TCR sequences (P1 and P2) published [28,30] are highlighted in grey boxes. The most dominant clones (D 1–3) within each specificity are highlighted in pink boxes.
(TIF)

**S9 Fig. GvHD monitoring.** (A) Enzyme levels in blood and (B) biomarker levels in plasma were used to monitor GvHD progress. GOT: glutamic oxaloacetic transaminase, AST: aspartate transaminase, GLDH: glutamate dehydrogenase, LDH: Lactate dehydrogenase, gGT: gamma-glutamyltransferase, CRP: C-reactive protein.
(TIF)

**S1 Table. Patient characteristics and treatment.** AITL: angioimmunoblastic T cell lymphoma; IgG: Immunoglobulin G; IgM: Immunoglobulin M; pos.: positive; neg.: negative; R-CHOP: Rituximab, Cyclophosphamide, Hydroxydaunomycin, Oncovin, and Prednisone; R-ICE: Rituximab, Ifosfamide, Carboplatin, and Etoposide; Fc: fragment crystallizable region; PD: progressive disease; SD: stable disease; allo-SCT: allogeneic stem cell transplantation; ATG: antithymocyte globulin; HLA: human leukocyte antigen; CMV: Cytomegalovirus; EBV: Epstein-Barr Virus; DLI: donor lymphocyte infusion; ATCT: adoptive T cell transfer; GvHD: Graft-versus-Host Disease; CsA: cyclosporin; HSV-1: Herpes-Simplex Virus-1.
(PDF)

**S2 Table. EPL-specific T cells.** TCRβ VJ ID: identification number for TCRβ variable-joining rearrangement, AA: amino acid.
(PDF)

**S3 Table. RAK-specific T cells.** TCRβ VJ ID: identification number for TCRβ variable-joining rearrangement, AA: amino acid.
(PDF)

**S4 Table. HPV-specific T cells.** TCRβ VJ ID: identification number for TCRβ variable-joining rearrangement, AA: amino acid.
(PDF)

**S5 Table. Presence of EPL-, RAK-, and HPV-specific T cells in different samples.**
(PDF)

## Acknowledgments

We would like to thank the staff of the department for transfusion medicine for their kind support of this study. Many thanks go to the staff of the Erlangen Bone Marrow Transplantation (BMT) unit and BMT outpatient clinic for excellent care taking of the patient.

## Author Contributions

**Conceptualization:** Volkhard Seitz, Andreas Mackensen, Armin Gerbitz.

**Data curation:** María Fernanda Lammoglia Cobo, Julia Ritter, Regina Gary, Volkhard Seitz, Josef Mautner, Michael Aigner, Simon Völkl, Stefanie Schaffer, Stephanie Moi, Anke Seegebarth, Kerstin Amann, Andreas Moosmann.

**Formal analysis:** María Fernanda Lammoglia Cobo, Julia Ritter, Regina Gary, Stefanie Schaffer, Stephanie Moi, Anke Seegebarth, Heiko Bruns, Kerstin Amann, Maike Büttner-Herold, Michael Hummel, Andreas Moosmann, Armin Gerbitz.

**Funding acquisition:** Volkhard Seitz, Andreas Mackensen, Armin Gerbitz.

**Investigation:** Anke Seegebarth, Heiko Bruns, Steffen Hennig, Armin Gerbitz.

**Methodology:** Steffen Hennig, Michael Hummel, Andreas Moosmann, Armin Gerbitz.

**Project administration:** Andreas Mackensen.

**Resources:** Heiko Bruns, Steffen Hennig.

**Software:** María Fernanda Lammoglia Cobo, Andreas Moosmann.

**Supervision:** Josef Mautner, Anke Seegebarth, Maike Büttner-Herold, Michael Hummel, Andreas Moosmann, Armin Gerbitz.

**Validation:** Josef Mautner.

**Visualization:** María Fernanda Lammoglia Cobo.

**Writing – original draft:** María Fernanda Lammoglia Cobo, Julia Ritter, Volkhard Seitz, Michael Aigner, Simon Völkl, Wolf Rösler, Maike Büttner-Herold, Michael Hummel, Andreas Moosmann, Armin Gerbitz.

**Writing – review & editing:** María Fernanda Lammoglia Cobo, Volkhard Seitz, Josef Mautner, Kerstin Amann, Maike Büttner-Herold, Steffen Hennig, Andreas Mackensen, Andreas Moosmann, Armin Gerbitz.

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
