## [Decision Letter · Decision Letter 0]

17 Jan 2022

Dear Lammoglia Cobo,

Thank you very much for submitting your manuscript "Reconstitution of T cell immunity against EBV in the immunocompromised host by adoptive transfer of peptide-stimulated T cells after allogeneic stem cell transplantation" for consideration at PLOS Pathogens. As with all papers reviewed by the journal, your manuscript was reviewed by members of the editorial board and by several independent reviewers. In light of the reviews (below this email), we would like to invite the resubmission of a significantly-revised version that takes into account the reviewers' comments.

We cannot make any decision about publication until we have seen the revised manuscript and your response to the reviewers' comments. Your revised manuscript is also likely to be sent to reviewers for further evaluation.

Sincerely,

Micah A Luftig, Ph.D.

Guest Editor

PLOS Pathogens

Erik Flemington

Section Editor

PLOS Pathogens

Kasturi Haldar

Editor-in-Chief

PLOS Pathogens

orcid.org/0000-0001-5065-158X

Michael Malim

Editor-in-Chief

PLOS Pathogens

orcid.org/0000-0002-7699-2064

Reviewer's Responses to Questions

**Part I - Summary**

Reviewer #1: Lammoglia Cobo et al. describe a case of AITL relapse soon after allogeneic hematopoietic cell transplantation treated with DLI, rituximab and ex vivo expanded EBV antigen-specific T cells. A particular focus was put on the characterization of the TCR clonotype content of the product and the monitoring of likely EBV-specific T cells post-transfer.

Such a strategy has been described several times, including the characterization of the product and immunomonitoring using TCR sequencing. However, the manufacturing method includes novel features (polyspecific antigen-specific CD8 T cell expansion guided by HLA allotypes) and the focus of the authors on the clonotypic characterization of the product and immune reconstitution offer interesting insights. Nonetheless, the authors should consider the following aspects to make their findings more impactful and their report more complete.

Reviewer #2: The authors demonstrate that Epstein Barr virus (EBV) derived peptide stimulation of peripheral blood mononuclear cells (PBMCs) derived from a patient with recurrent angioimmunoblastic T cell lymphoma (AITL) after transplantation leads to the expansion of EBV specific CD8+ T cells. These are primarily directed against peptides from BZLF1 and EBNA1. Particularly EBNA1 specific CD8+ T cells were dominated by one TCR clonotype. Several of the identified EBV specific TCR clonotypes expand after transfer into the patient and are maintained for at least 8 months. From these data the authors conclude that a diverse set of EBV specific clonotypes can be expanded with their protocol and transferred for treatment of AITL, reestablishing EBV specific immune control after stem cell transplantation.

Even so the authors apply TCR clonotype tracing to follow adoptively transferred EBV specific T cells in one bone marrow transplant patient for the first time, it is unclear what new insights they gain from these studies. Persistence of such adoptively transferred T cells has previously been documented for up to 18 months (Heslop et al., Nat Med 1996). Therefore, the authors should address in more detail how the adoptively transferred T cell products control the pathogenic T cell expansion in this patient and if there are differences between the detected EBV specific TCR clonotypes.

**Part II – Major Issues: Key Experiments Required for Acceptance**

Reviewer #1: Major points:

1- The expanded T cells were directed against antigens from BZLF1, EBNA3A and EBNA1. In vivo, BZLF1-specific T cells expanded more that the EBNA-1 specific T cells (which was not the case ex vivo). Did the AITL at relapse (or at diagnosis) express the targeted proteins? Not all AITL express latency type 3 proteins and, as such, the AITL expression of BZLF1 should be confirmed (especially in a context in which AITL relapse clearly appears to be the cause of the rising EBV viral load)? Otherwise, what can account for this difference (exhaustion markers expression, poor functional features, immunodominance, etc)? Additional characterization of the product along these lines would improve the manuscript.

2- The clonotype composition of sorted cells did show significant overlap between Ag-specific populations. Although resolved by the application of “filters,” doubts remain regarding the true Ag-specificity of the sorted cells and whether sorting was stringent enough. Functional (cytokine secretion, degranulation) tests using sorted cells in the presence of their target peptides vs non-target peptides would convince further.

Reviewer #2: 1. The composition of the uncontrolled EBV infection in the investigated patient should be characterized in more detail. Was EBV detected in the patient’s AITL cells? Were there also elevated EBV titers in the CD4+ T cell negative fraction?

2. The authors report the contribution of several TCR clonotypes to the T cell response against BZLF1 and EBNA1 specific T cell reactivities. What is the difference between the detected TCR clonotypes? Do the authors have any evidence that some of these were of higher affinity for the identified epitopes? Did this correlate with their expansion in vitro or in vivo?

3. Did certain TCR clonotypes recognize the autologous AITL cells more efficiently than others? Was there any difference in EBNA1 versus BZLF1 specific T cell recognition of autologous AITL, if this actually harbors EBV?

**Part III – Minor Issues: Editorial and Data Presentation Modifications**

Reviewer #1: Minor Points:

1- The description of the case. The report is at times confusing – to help with the flow of the manuscript, the content of Table S1 should be described at the beginning and a timeline provided (time from chemotherapy to transplant unclear, various treatments, etc) in the body of the manuscript.

2- The occurrence of severe GVH after adoptive transfer should be discussed (role of DLI, ATCT?, impact of steroid-therapy on reconstitution and in this case, the occurrence of a fatal infectious outcome). Likewise, DLI and Rituximab seem to have had a greater therapeutic impact than ATCT (not clear that it was administered in the context of a rising EBV viral load). The potential implications of that should be discussed as well.

3- Fig 4 shows that the donor PBMCs contain several of the abundant product clonotypes. Although the author provide a rationale to link ATCT to EBV-specific immune reconstitution, they do not rule out the possibility that clonotypes in the DLI contributed to the immune reconstitution. This should be mentioned and discussed.

4- The EBV viral load measurement method is not specified – plasma/serum or whole blood? Commercial assay or LDT?

5- As opposed to what is described in the text, Fig S1 does not show data on IFNg.

Reviewer #2: 1. The title should indicate that the study reports only one patient and that this patient suffered from AITL.

PLOS authors have the option to publish the peer review history of their article (what does this mean?). If published, this will include your full peer review and any attached files.

Reviewer #1: No

Reviewer #2: No
---

## [Editor Report · Decision Letter 1]

31 Mar 2022

Dear Lammoglia Cobo,

We are pleased to inform you that your manuscript 'Reconstitution of EBV-directed T cell immunity by adoptive transfer of peptide-stimulated T cells in a patient after allogeneic stem cell transplantation for AITL' has been provisionally accepted for publication in PLOS Pathogens.

Best regards,

Micah A Luftig, Ph.D.

Guest Editor

PLOS Pathogens

Erik Flemington

Section Editor

PLOS Pathogens

Kasturi Haldar

Editor-in-Chief

PLOS Pathogens

orcid.org/0000-0001-5065-158X

Michael Malim

Editor-in-Chief

PLOS Pathogens

orcid.org/0000-0002-7699-2064
---

## [Editor Report · Acceptance letter]

13 Apr 2022

Dear Mrs. Lammoglia Cobo,

We are delighted to inform you that your manuscript, "Reconstitution of EBV-directed T cell immunity by adoptive transfer of peptide-stimulated T cells in a patient after allogeneic stem cell transplantation for AITL," has been formally accepted for publication in PLOS Pathogens.

Best regards,

Kasturi Haldar

Editor-in-Chief

PLOS Pathogens

orcid.org/0000-0001-5065-158X

Michael Malim

Editor-in-Chief

PLOS Pathogens

orcid.org/0000-0002-7699-2064